# The Vice of Social Comparison in Kierkegaard: Nature, Religious Moral Psychology, and Normativity

## Wojciech Kaftanski

Interdisciplinary Centre for Ethics, Faculty of Philosophy, Jagiellonian University, Grodzka 52, 31-044 Kraków, Poland; wojciech.kaftanski@uj.edu.pl

**Abstract:** This paper argues for the thesis that social comparison is, for Kierkegaard, a vice. The first part of this article reconstructs Kierkegaard's understanding of the nature of social comparison. Here, I bring attention to his anthropological but also political and sociological observations that pertain to social comparison and its links to modernity. The second part reconstructs the moral psychological account of social comparison in Kierkegaard, drawing on some of the available secondary literature. I complement Kierkegaard's consideration of social comparison in relation to worry and humility with his account of the non-cognitive aspects of its operationality. The third part demonstrates that social comparison is a vice. Therein, drawing on the previous sections of this article, I identify Kierkegaard's naturalistic argument engaged to present social comparison as a non-moral and non-religious vice (functionalism), pointing toward its intermeshing with the moral religious.

**Keywords:** social comparison; vice; virtue; virtue ethics; Kierkegaard; ethics

---

"Spirit is precisely: not to be like the others". (M: 344)[1]
"Social comparisons have both cognitive and affective consequences". (Gilbert et al. 1995)

## 1. Introduction

*Kierkegaard After MacIntyre* has opened an important avenue for reading Kierkegaard in dialogue with virtue ethics. This and much of the ensuing and contemporary scholarship explore the role of virtues in the transition from the aesthetic to the ethical form of life (Rudd 2001; Davenport 2001a; Lillegard 2001), but also the nature of particular virtues and the relationships between them (Lippitt 2020). While skepticism concerning reading Kierkegaard's works in relation to virtue ethics has been articulated (Walsh 2018), the dominant positive view of virtues in Kierkegaard is that their underpinnings are theological (Vos 2016, 2020) and teleological (Rudd 2001; Davenport 2001b) in nature.

These positive readings seem to depart from the classical Aristotelian–Thomistic framework, wherein virtues operate on resources available in the composition of human nature to accomplish eudaimonia. The main point of contention pertains to the metaphysical foundation of ethical concern for Kierkegaard, for whom virtues (but also duties and obligations) are key to achieving the "eternal happiness" of being with God. It seems that the ideal of a good life falls short of Kierkegaard's ultimate interest in perfecting one's character, which in turn opens an individual to the realm of the religious sphere of life. Moreover, John Davenport's "Towards an Existential Virtue Ethics: Kierkegaard and MacIntyre" from the cited volume makes a case for Kierkegaard's existential virtue ethics, suggesting that rather than the goal of eudaimonia, a virtuous character in a Kierkegaardian sense should strive to be authentic rather than "happy". The ideal of authenticity, which Davenport defines in phronetic terms as "practical coherence among earnestly willed projects that can give narrative shape and enduring meaning to a human life" (Davenport 2001b, p. 265), emphasizes freedom, will, and individuality at the expense of habit and sociality. Davenport notices, following Hannay, that Kierkegaard's ethical program is not founded in universal human teleology, but rather that it points to the realm of the transcendent God as

the guarantor of ethics. "This remains true at the religious stage: our *telos* understood as the happiness of salvation does not *ground* the virtues of moral life or faith for Kierkegaard but is instead their final complement or true reward" (Davenport 2001b, p. 275). Davenport interprets Kierkegaard's ethical engagement with virtues as a project that is really picking up the fragments of the Eudaimonist tradition in the wake of the Reformation. The goal of his new synthesis is to reconstruct on a new basis what was valuable in eudaimonism, while recognizing that any acceptable understanding of free will must render untenable a fundamental claim of the primary Aristotelian version of that tradition (Davenport 2001b, p. 287).

Obviously, such an ethical program can be completed fully only in "a religious context of faith" for Kierkegaard; however, it does not build on the religious, but it rather leads to it.

In this article, I take Kierkegaard's remarks on the human propensity to compare oneself with others, which I term, following substantial psychological and sociological literature and also Kierkegaard's own lexicon ("worldly life of comparisons"; UDVS: 189), "social comparison", to be a basis of considering it a vice in three senses: non-moral/non-religious, moral, and religious. I will demonstrate that while social comparison is for Kierkegaard a problem on these three levels, it is the comparison that is first and foremost non-moral and non-religious that informs the moral and the religious. Put differently, I show that it is important to understand how social comparison is articulated by Kierkegaard in largely psychological and sociological terms to better understand how it is detrimental to our moral and spiritual lives. In this exposition I follow Davenport's "existential virtue ethics", which he rightfully identifies in Kierkegaard; yet, I will bring more attention to reasons why comparison is detrimental to us in the sense of our general well-being, not just to our ethical-religious lives. While he acknowledges the eudaimonic inspirations in Kierkegaard's ethics of virtues, Davenport seems to downplay them on account of phronetic authenticity. My reading of Kierkegaard's account of social comparison will investigate the nature of social comparison vis a vis human nature and its teleology to demonstrate that social comparison is bad for our well-being in a general sense close to the eudaimonic (flourishing) tradition. It is the dispositional dimension of social comparison that Kierkegaard finds to be decidedly vicious. Our lives go badly when we tend to compare ourselves with others, when we have or cultivate in ourselves such comparing dispositions that constitute in us a life of continuous and compulsive social comparison. Social comparison modifies many of our dispositions and concerns that, while they may have noble motivation, are being unwittingly transformed to vicious dispositions. Acts of social comparison are bad because they result from our character that is prone to social comparison and because they shape it by creating or reinforcing its comparison tendencies. While it is in our nature to compare with others, social comparison in Kierkegaard is presented as that which goes against our nature when abused or used in unwarranted contexts; hence, it goes against what Aristotle would understand as the proper function of the human being.

The first part of this article reconstructs Kierkegaard's understanding of the nature of social comparison. Here, I bring attention to his anthropological but also political and sociological observations that pertain to social comparison and its links to modernity. The second part reconstructs the moral psychological account of social comparison in Kierkegaard, drawing on some of the available secondary literature. I complement Kierkegaard's consideration of social comparison in relation to worry and humility with his account of the non-cognitive aspects of its operationality. The third part demonstrates that social comparison is a vice. Therein, drawing on the previous sections of this article, I identify Kierkegaard's naturalistic argument engaged to present social comparison as non-moral and non-religious vice (functionalism), pointing toward its intermeshing with the moral-religious.

## 2. Social Comparison: Nature

Aristotle's virtue ethics is primarily focused on the individual dimension of human character. The subjects of character formation are the citizens of the polis, a "chosen few", often with privileged upbringing, financial security, and plenty of leisure time. Non-citizens' participation in social life and their influence on the shape of social values and mores are rather limited in Aristotle's time.

Kierkegaard writes in a significantly different historical context, one that is fundamentally determined by the institutions of the Christian churches and Christian culture, where philosophical reflection is situated in a large-scale social-political project of the Enlightenment. Kierkegaard's engagement with virtue ethics attempts to integrate the problem of the will prevalent in the Christian tradition of philosophy from the Middle Ages to the Reformation (Rudd 1997; Davenport 2001b; Fremstedal 2022; Vos 2020), but it also attempts to accommodate a new kind of human experience, the experience of modernity (Kirmmse 1990; Pattison 2013; Stewart 2015; Kaftanski 2022). Situating the eudaimonist tradition of virtue ethics in a new social context, Kierkegaard reinterprets old ("sin") and "introduces" new ("despair" and "anxiety") phenomena to an ethical landscape providing us with insightful, but not infrequently convoluted, moral psychological descriptions (Hanson 2022). The new social context requires the identification of a new set of virtues that should be cultivated by a morally good person, and a new set of vices that a good person needs to avoid.

One such vice that I identify in Kierkegaard's writings is social comparison. He sees social comparison to be a natural human tendency and a capacity to compare with others. He recognizes modernity as especially augmenting our comparing dispositions and acts. Witnessing the advent of a sociological reflection in the modern sense, Kierkegaard often attributes modernity with predicates, which, while obsolete today ("a prodigious monstrosity with many heads" (WA: 229)), point toward his interest in understanding social complexities. What many early sociologists perceived naively as the aim of human interactions (advancing collectivity) and the unavoidable and positive consequence of increased human sociality (deindividualization and uniformity), Kierkegaard observes with increased suspicion and criticism. While Kierkegaard attributes various negative modern phenomena with a religious dimension, it is important to recognize their natural underpinnings featured in his work.

Social comparison is at the heart of the adverse implications of modernity on the quality of human life. It is essentially tied to the modern phenomenon of mass congregation that creates a favorable environment for cultivating social comparison. Social comparison is, as I will argue, a vice of which the most problematic aspects are being activated in collective society. Human collectivity requires an effort of the adaptation of social rules and mechanisms oriented toward alleviating social tensions through seeking points of commonality and promoting uniformity.

> "Modernity is characterized by a dissolution of differences leading to normative uniformity on an unprecedented scale. The activities of the masses amount to spontaneous reflexive processes produced by mimetic re-actions between anonymous people assembled in gatherings—"the crowd"—but also assembled in groupings that are not characterized by physical proximity—"the public". Significantly amplified by modernity, human sociability and collectivity have impacted people on the individual level. Kierkegaard sees this trend as effectively aiming at challenging and, as a result, marginalizing the value of human individuality, the meaning of subjective experience, and the role of passion and faith in daily life. Modernity forces individuals to rethink their sense of identity, their place in the world, and the meaning of life". (Kaftanski 2022, p. 1)

It is precisely the crisis of individuality, identity, and the meaning of life caused by the decline of the modern institutions and the influence of sociality on the quality and orientation of human existence that Leon Festinger identifies at the core of our motivation

to compare ourselves with others in modernity in his (Festinger 1954) "A Theory of Social Comparison Processes". For Festinger, there is a strong correlation between what he calls "an objective, non-social basis for the evaluation of one's ability or opinion" and our tendency to "evaluate [our] opinions or abilities by comparison with others" (Festinger 1954, p. 120). More specifically, our disbelief in impartial benchmarking for moral judgement or moral performance is correlated with our increased comparing with others. The fact that we are of the correct opinion about various dimensions of our lives has direct consequences on the quality of our lives. Misapprehension may cost us our social status, jobs, and, in some cases, our health. While we can, in some cases, evaluate our ability in relation to the physical world (objective model), many such important aspects of our lives such as social norms require a different object for evaluation; in such cases, we rely on the opinions of others (proxy model).

While it is indeed modernity that has dramatically amplified our social comparison dispositions, it only brought to light something that is part of our makeup. Comparison is a driver of social adaptation and cooperation. We are equipped with "a drive to determine whether or not" our opinions and behaviors are "correct" (Festinger 1954, p. 118). Comparison is an advantage essential to human survival; it lays at the foundation of human culture. "Thinking without comparison is unthinkable. And, in the absence of comparison, so is all scientific thought and scientific research" (Swanson 1971, p. 145). Comparing is part of our daily practice. It is a critical capacity that allows us to distinguish between similar objects with respect to their modalities such as substance, magnitude, and relation, and also between objects that are quite different such as water, music, and human behavior (Kruglanski and Mayseless 1990). The skill of comparison also allows us to evaluate such complex objects as ideas, values, numbers, and words. In our daily life, we compare products and prices. No one wants to overpay for their car, apartment, clothing, or daily shopping. We want the best value for our money and efforts.

Not all comparison is bad, and, arguably, not all social comparison is bad either. Rather, the point I am making here is that the advantage derived from social comparison turns against us when we engage it in (challenging) situations that prompt us to build our character. Just as fear pathologically morphs into anxiety when activated unwarrantedly, social comparison becomes detrimental to our well-being when we engage in it indiscriminately in all contexts. It is first and foremost our tendency to compare ourselves with others that Kierkegaard criticizes, but he admits that social comparison is amenable to situational awareness and considerations of degree. These aspects of social comparison make it responsive to virtue ethics more than deontological or utilitarian ethical frameworks. Virtue ethics help us devise mitigating strategies when dealing with social comparison disposition; they also disclose the moral psychological complexity of social comparison that warns us against the perils of battling social comparison, which may perversely lead to reinforcing them in areas that affect us below the cognitive register.

Kierkegaard finds social comparison at the foundation of human sociality in its historic sense. It is at work in individual human affairs but also plays an important part in the historic development of humanity. In "How Glorious It Is to Be a Human Being" from *Upbuilding Discourses in Various Spirits*, Kierkegaard says thus:

Alas, those great, uplifting, simple thoughts, those first thoughts, are more and more forgotten, perhaps entirely forgotten in the weekday and worldly life of comparisons. The one human being compares himself with others, the one generation compares itself with the other, and thus the heaped-up pile of comparisons overwhelms a person. As the ingenuity and busyness increase, there come to be more and more in each generation who slavishly work a whole lifetime far down in the low underground regions of comparisons. Indeed, just as miners never see the light of day, so these unhappy people never come to see the light: those uplifting, simple thoughts, those first thoughts about how glorious it is to be a human being. And up there in the higher regions of comparison, smiling vanity plays its false game and deceives the happy ones so that they receive no impression from those lofty, simple thoughts, those first thoughts (UDVS: 189).

This complex account of a life of comparison, with a Hegelian angle to it, presents a link between the human tendency to compare with others and what Kierkegaard renders as a generational comparison that has an impact on the formation of social classes. Comparison, while oriented toward annihilating differences between people, furthers social distinction and "predispose(s) them to social and political violence" (Pattison 2013, pp. 21–22). Increased self-preoccupation, a hallmark of modernity, divides the society in two classes: those who are worse-off and better-off. The worse-off are those that occupy "the low underground regions of comparisons"—they are "unhappy people". The better-off "in the higher regions of comparison" are happy ones. This distinction between unhappy and happy people maps out on the distinction in Social Comparison Theory that identifies downward and upward types of comparison. The upward comparison is when someone who is perceived to be better-off compares with someone perceived as worse-off; analogously, the worse-off comparing with the better-off is upward comparison (Wood 1996).

While I will discuss in more detail the moral psychological and spiritual dimensions of social comparison in Kierkegaard, it is important to notice in the quoted passage that regardless of the type of comparison one engages, it is problematic to their well-being in four senses.[2] First, comparison has a propensity to accumulate to the degree that it eventually "overwhelms a person". Second, comparison takes away the animating and enlivening experience of appreciating being a human being. In both cases, the subjective experience of well-being is compromised. This leads us to being unhappy, which is also the case for the seemingly happy person who is in fact deceived about their state of unhappiness (Cf. SUD: 25). Three, comparison is not simply a mental act. Its very nature is about comparison becoming a fossilized disposition that becomes a type of life, a lifestyle, which Kierkegaard calls "worldly life of comparisons" (UDVS: 189). Four, in the nature of social comparison is the fact that our comparing with other people is not a private matter but one that has a generational influence on social interactions. This generational aspect speaks to the deterioration of social relations, which contrasts with the Hegelian position about the increase of individual, and especially collective, societal life. Kierkegaard recognizes that it is in our nature to compare ourselves with others and that it is in the nature of modernity to increase social comparison dispositions in us that will increase the negative consequences on our individual and social lives, making our lives increasingly worse with every new generation.

The titles of two essays from *Upbuilding Discourses in Various Spirits* wherein Kierkegaard offers a more in-depth analysis of social comparison as a phenomenon that helps us understand it in its own right are: "To Be Contented with Being a Human Being" and "How Glorious It Is to Be a Human Being". Kierkegaard's location of social comparison in the context of his discussion of human nature suggests that this phenomenon is central to what we are in two ways. On the one hand, speaking of social comparison, Kierkegaard refers to a necessity to properly understand our functioning in the context of a more general human anthropology. He identifies that which is conducive to our well-being in the sense of existentially qualified categories of happiness and authenticity. On the other hand, Kierkegaard identifies social comparison as that which is conducive to us acting against our nature, frustrating our teleology. In the Aristotelian sense, Kierkegaard seeks to determine why social comparison negatively affects our well-being understood in the functionalist sense of flourishing (Shields 1991). In both senses, social comparison motivates us to act against our understanding of and our contentment with "Being a Human Being".

## 3. Social Comparison: Religious Moral Psychology

Not all comparison is bad. In fact, it is often recommended by Kierkegaard. The above-quoted passage from *Upbuilding Discourses in Various Spirits* follows from Kierkegaard's advice that one should compare themselves with the lily of the field: "But by looking at the lily the worried one is reminded to compare his clothing with the lily's—even if poverty has clothed him in rags" (UDVS: 189). Elsewhere in that volume, the author recommends

comparing one's endured suffering with the prospect of life in eternity to gain a beneficial perspective on suffering. Such comparison helps us bear adversity: "But the comparison does not weigh in such a manner that it keeps the two magnitudes far apart from each other; instead, it brings them so close together that the presence of the eternal happiness changes the expression about the hardship" (UDVS: 315). Some forms of comparison that Kierkegaard admits have a social and moral-religious dimension. Throughout his *Journals*, Kierkegaard states that Christ, "the God-man as the prototype", is "the criterion for being human", and we should make conscious comparisons with Him to understand how far we are from the ideal human being represented by Christ (JP 2: 1802; JP 1: 334; JP 2: 1895). We are also invited to compare between people to have a good grasp of important qualities that are religious and non-religious, such as dishonesty, mediocrity, but also genius (Cf. JP 3: 3088; JP6: 6553; JP6: 6560).[3] Rather than comparing oneself with particular individuals or groups, we should compare and conform our lives to the standard of existence that is "all men" (JP3: 2966).

Kierkegaard's criticism of social comparison scrutinizes both the phenomenon and the emotions and dispositions that either produce or augment it, are caused or amplified by it, or are linked to it. John Lippitt's "Beyond Worry? On Learning Humility from the Lilies and the Birds" links social comparison with worrying and argues that it is mitigated by humility. As Lippitt notices, for Kierkegaard "'all worldly worry' is based upon an 'unwillingness to be contented with being a human being'" (Lippitt 2019, p. 91; UDVS: 178–179). Behind this attitude of unwillingness is social comparison rendered as an agitating and bewitching "voice" that continuously convinces us to fall for "diversity", which for Kierkegaard is a form of otherness harmful to human nature. Social comparison is "the seductive in the human being" and "the restless mentality of comparison, which roams far and wide, fitfully and capriciously, and gleans the morbid knowledge of diversity" (UDVS: 169). On the other hand, "the spirit of comparison" is the cause of our inability to rely on God manifested in our overt preparedness. We do not want to be caught off guard; we plan for emergencies because we worry. Lippitt convincingly identifies humility in Kierkegaard as the trait that allows us to manage our social comparison disposition, as he suggests that humility is not about thinking less *of* oneself (as this, in fact, is based on comparing oneself with others by ranking oneself lower), but less *about* oneself. To combat the spirit of comparison we must be less inward-looking and more outward-looking; less about ourselves, more about other people. Comparison can be beneficial to us, argues Lippitt, but the one comparing oneself with an exemplary figure (upward comparison) must control one's "competitive ego". Additionally, following Robert C. Roberts, Lippitt (2019, pp. 96–97) suggests that social comparison can be substantially reduced, if not eliminated, if we cultivate "a transcendent form of self-confidence", "the radical self-confidence that Christians call humility: a self-confidence so deep, a personal integration so strong that all comparison with other people, both advantageous and disadvantageous, slides right off him" (Roberts 2007, p. 90).

These diagnostic and prescriptive perspectives on social comparison are invaluable. They are well-argued and beneficial to individuals struggling with social comparison. They emphasize the role of our cognitive capacity to control and curtail social comparison. I would like to complement them with an account of a dimension of social comparison that falls beneath the radar of our cognitive register that I find in Kierkegaard's writings. More specifically, my goal is to unearth Kierkegaard's view of social comparison that has an affective and imitative dimension. It rests on an etymological analysis of the word "comparison" in Danish that Kierkegaard uses and on his category of difference.

The Danish for "comparison" is *Sammenligning*, which contains the word *ligne*, which means relational likening, such as between objects. The objects that are alike are comparative. *Sammenligning* indicates imitative likening of oneself with others. It invites various degrees of similarity. There is a greater similarity between individual human beings than between humans and chimpanzees and other species. But humans are not completely alike; there are differences between them. Social comparison, I argue, is a category from Kierkegaard's moral psychology that accounts for the human reflective (see Lippitt) and

affective imitative inclination to look for a relative point of reference to appraise a norm, value, or situation.

We are alike but also different. In one of his final writings, Kierkegaard writes: "Spirit is precisely: not to be like the others" (M: 344). Yet, our seeking to not be like others invites qualification that should clarify what that really means and how that can be achieved without falling back on social comparison. In that respect, Kierkegaard introduces two important concepts. Difference is key to distinguishing between individual human beings. Difference (*Forskjellen*), and related to it, dissimilarity (*Forskjellighed*), are for Kierkegaard moral-religious categories. In *Works of Love*, difference is presented as a positive category, while dissimilarity, or more specifically "the dissimilarity of earthly life", is presented as a negative category. Difference pertains to our essential individuality ("That is how high Christianity has placed everyone, unconditionally every human being, because for Christ, as for God's providence, there is no number, no crowd; for him the countless are counted, are all individuals" (WL: 69)). Dissimilarity pertains to our earthly differences of station, financial situation, and physical and mental capacities—as Kierkegaard says, "none of us is pure humanity" (WL: 70). While we should seek difference, we must ascertain that we avoid two extremes: building our self-worth on earthly dissimilarities and building our self-worth on the ideal of their eradication. Kierkegaard writes:

> "Thus, Christianity has once and for all banished that abomination of paganism, but the dissimilarity of earthly life it has not taken away. This must continue as long as temporality continues and must continue to tempt every human being who comes into the world, inasmuch as by being a Christian he does not become exempt from dissimilarity, but by overcoming the temptation of dissimilarity he becomes a Christian". (WL: 70)

Taking advantage of eradicating dissimilarity gets us back to comparison, which Kierkegaard calls "comparing dissimilarity".

Dissimilarity is like an enormous net in which temporality is held; there are in turn variations in the meshes of this net—one person seems more trapped and bound in existence than another; but all this dissimilarity, the dissimilarity between difference and difference, this comparing dissimilarity, does not preoccupy Christianity at all, not in the least—such a preoccupation and concern is again nothing but worldliness.

Kierkegaard wants us to avoid three positions here: engaging what he calls "comparing dissimilarity" (WL: 71), seeking feelings of superiority when we compare with the worse-off, and engaging social comparison unknowingly or unwillingly.

These three positions that we should avoid we find in his 1848 discourse: "The Tax Collector" from *Three Discourses at the Communion on Fridays*. Analyzing Luke 18's "The Parable of the Pharisee and the Tax Collector", Kierkegaard invites us to consider the insidious power of social comparison that penetrates our social-ethical and religious lives. Originally, this parable is directed to "some who were confident of their own righteousness and looked down on everyone else" (NIV 18: 9). Kierkegaard's motivation to engage this story is to tackle social comparison head on. While worry is not central to the parable (yet arguably compatible), and humility is represented in light of its opposite, hypocrisy, it is the nature of social comparison that corrupts us here.

The parable presents the Pharisee and the tax collector entering the temple to pray to their God. It is the tax collector, a figure generally despised by his contemporaries for extorting taxes from them and collaborating with the occupying administration of the Roman Empire, that "went home justified before God". It is not the Pharisee, a widely respected figure in Jesus' times for their high moral standards and piety. This is so because, as the story tells, the Pharisee justified himself before God. This self-justification is prompted by his usage of "the criterion of human comparison" (WA: 129). The Pharisee takes the tax collector to be morally and spiritually worse-off. He looks down on him, and this produces in him a positive feeling about himself, which perhaps reinforces his self-righteousness. He is "the hypocrite who deceives himself and wants to deceive God" (WA: 127). The tax collector is "standing by himself" and "staying far away". He recognizes that in God's eyes

the difference between him and the Pharisee is meaningless, yet this fact lessens neither his guilt nor his awareness of his moral failings. Realizing that the Pharisee is the bad character in the told parable requires minimal effort from Kierkegaard's contemporaries, but also from contemporary readers. We, the readers of the story, naturally identify with the tax collector, who exhibits moral-spiritual superiority over the Pharisee at the temple. We, as Adam Smith would say, sympathetically identify with him (Smith 2002, p. 21). Sadly, we do so, as David Hume would say, to seek pleasure from comparing with the worse-off, the Pharisee. The readers

> "resemble the Pharisee [*der ligne Pharisæeren*] but have chosen the tax collector as their prototype, hypocrites who "trust in themselves that they are righteous and despise others," while they nevertheless fashion their character in the likeness of the tax collector and sanctimoniously stand far off, unlike the Pharisee, who proudly stood by himself; sanctimoniously cast their eyes to the earth, unlike the Pharisee, who proudly lifted his eyes to heaven; sanctimoniously sigh, "God, be merciful to me, a sinner," unlike the Pharisee, who proudly thanked God that he was righteous. (WA: 127)

Kierkegaard draws our attention to a dissonance between our conscious "choosing" of the tax collector as our model and our unconscious and perhaps unwilled adaptation of the model of the Pharisee. This apparent discord between the cognitive and non-cognitive helps us understand the affective dimension of social comparison. Being an adaptive mechanism that is part of our survival toolkit, social comparison has the potential to be activated unbeknownst to us. The readers may truly understand the moral superiority of the tax collector and identify with him, but their acts—embodied and mental—speak to the fact that they have modeled their behavior after the Pharisee, saying: "God, I thank you that I am not like this Pharisee" (WA: 127).

Social comparison produces in us various emotions that can be pleasurable or painful. Following Hume, we feel pleasure in taking pride in our possessions and qualities. Comparison with the worse-off increases pleasure; comparison with the better-off gives us pain. We are naturally predisposed to seek to compare with the worse-off to increase the feeling of pleasure (James 2005; Hartmann 2021). Indeed, "The Pharisee proudly found satisfaction in seeing the tax collector" (WA: 130). Kierkegaard seems to be warning us to avoid cultivating in us parasitic uplifting feelings that either produce or rely on contempt for others and our feeling of self-righteousness.

While reflection can help curtail social comparison, reflection can also stimulate it. In *Works of Love*, Kierkegaard argues that reflection negatively affects love when it engages comparison for its evaluation. He draws on the argument on the nature of love and its incommensurability with any reflection oriented toward discriminating degrees of love. There is one love that is "eternally secured" by God being love (WL: 34), and any attempt to determine it by comparing manifestations of love is futile. Put differently, Kierkegaard does not mean here that we should never question whether we are in love or are being loved (we can indeed be in abusive relations that are far removed from love). Rather, his argument is that the nature of love determines that we cannot define it by comparing acts of love, which Kierkegaard calls "testing". Two factors are identified in "testing:" our inclination to compare and our anxious nature. "The testing undoubtedly has its basis in love, but this violently flaming desire to test, this craving desire to be put to the test, denotes that the love itself is unconsciously uncertain" (WL: 33). It is in our nature to put things to the test, which includes ourselves and others. This inclination is exacerbated by anxiety, which brings to light our nervousness about love. "This security of eternity casts out all anxiety and makes love perfect, perfectly secured" (WL: 32). Anxiety is likewise a human condition, one that Kierkegaard relates in *The Concept of Anxiety* to our sinful nature. These two inclinations combined are a dangerous concoction that is harmful to our well-being because it affects how we relate to ourselves, others, and the world.

## 4. Social Comparison: Vice

In the previous sections, I lay out the nature of social comparison and its religious moral psychology in Kierkegaard. We learn from Kierkegaard that social comparison is part of our nature; it is engaged individually and collectively by humans. It is associated with various emotions. And various emotions are produced by it and also produce it. Social comparison has a cognitive and non-cognitive dimension.

Social comparison is problematic to our well-being. Due to its complex nature, we cannot and should not completely eradicate it. I find it useful to tackle the normativity around social comparison by considering it as a vice according to the tradition of virtue ethics. This moral tradition seems more amenable to the evaluation of social comparison as it is not focused on its eradication or regulation according to a maxim (deontological traditions), and it does not seek the maximization of pleasure and the minimization of pain caused by social comparison (utilitarian and consequentialist traditions). Virtue ethics seeks to determine whether social comparison is good for us in the sense of our general well-being.

As I have already indicated, Kierkegaard's engagement with virtue ethics is unique in the sense of interrelating eudaimonism and authenticity as the goals and indicators of being well. Social comparison is a vice for Kierkegaard because it makes us inauthentic and unhappy. Unregulated by the will, when engaged indiscriminately, social comparison dominates our lives to the extent that we become governed by the urge to compare ourselves with others, real and imagined. Unregulated by the will, and unscrutinized by reason, social comparison parasitically clings to our noble motivations—for instance, it has the power to alter the trait of humility into hypocrisy. "The idolized positive principle of sociality in our age is the consuming, demoralizing principle that in the thralldom of reflection transforms even virtues into *vitia splendida* [glittering vices]" (TA: 86). Kierkegaard's tax collector discourse shows that social comparison affects in us a double mindedness when we willingly take non-comparing figures—such as the tax collector—as our role models, but the mechanism behind this choice is, sadly, based on social comparison.

Social comparison is a vice that gives comparative direction to our lives, which amounts to what Kierkegaard would call a comparative life-view. To have a life-view is, for Kierkegaard, to have a positive idea or a set of values that structures one's existence. "A life-view provides an individual with a unified, hence non-fragmented vision of life, and of life's teleology, meaning, and purpose. Furthermore, a life-view grounds an individual in the actual" (Kaftanski 2022, p. 61). A life of comparison is directed by our comparison perspective on the world.[4] Kierkegaard describes a person with the life view of social comparison as someone who "looks comparingly at others" (UDVS: 179). The life-view of social comparison creates an environment that is conducive to fostering a life rife with dispositions and attitudes that are harmful to our sense of identity and selfhood. Such a life is a fertile ground for negative emotions and dispositions that make us inauthentic.

Being the root cause of worrying, social comparison causes us to worry and builds in us insecurity on three levels: the insecurity related to the fulfillment of our real needs; the insecurity caused by seeing that others have greater relative security; anxious insecurity that spirals out of control and seeks and often imagines objects that can be compared. This last form of comparison abstracts from human-to-human interactions and projects worrying about the world in a more general sense: "He anxiously compares one day with another, if on the day he has rich abundance he says: But tomorrow! and if on the day he feels the pinch of scarcity he says: Tomorrow will be even worse—then he certainly is making comparisons" (UDVS: 179).

An Aristotelian perspective on Kierkegaard's critical appraisal of social comparison discloses reasons for considering this phenomenon as being problematic in the sense of it harming our overall well-being. The argument that I am putting forward is that Kierkegaard considers social comparison to be harmful to our acting that agrees with and responds to particulars of our human nature. Comparing ourselves with others, in the sense I have presented, is against our interest and contrary to who we are as selves that live

up to their full potential. Reasons for that are spread throughout Kierkegaard's authorship. In *The Sickness unto Death*, those who do not want to be themselves and want to be others are in despair. A despairing self is one that is plagued by misrelations, which are improper relations to oneself and others, including God.

The perilous aspect of one's desire to be like others is examined by Kierkegaard in the parable of "the worried lily" (UDVS: 167–171). This widely commented-upon passage from *Upbuilding Discourses in Various Spirits* presents a lily, which while initially "joyful and free of care", becomes infatuated with a bird that convinces the lily to act against its nature and causes its demise (Cf. Carron 2019). Kierkegaard uses this story to demonstrate that social comparison is dangerous to us, and the principle of harm is in acting against one's nature. The bird's nature is to "not remain in the same place", while the lily by nature remains in one place "just like the flowers". Even though the difference in natures "struck the lily as odd and inexplicable", it acquired longing for qualities foreign to its nature such as freedom of movement. The story insists that it is the insidious nature of the bird and the affective and imitative nature of the acquisition of desire that contribute to the lily's death. The bird keeps tempting and luring the lily by offering stories that are both true and false ("true and untrue, fiction and truth" (UDVS: 169)); it makes sure that the lily's self-worth is diminished after each visit: "It usually ended its story with the comment, humiliating to the lily, that in comparison with that kind of glory the lily looked like nothing—indeed, it was so insignificant that it was a question whether the lily actually had a right to be called a lily" (UDVS: 167).

The lily experiences several objectively detrimental effects of the encounter with the bird that speak to its diminished mental and physical well-being. They do not have any specific moral or spiritual underpinnings: the lily becomes worried, is deprived of a good sleep, wakes up restless, experiences boredom, and suffers from a lack of self-worth and good mood. These symptoms belong to generally acknowledged signs of anxiety or mild depression (Roelofs et al. 2008).[5] In fact, the train of thought and the internal dialogue of the lily reminds us of the phenomenon of racing or ruminative thoughts—fast, repetitive thought patterns about a given subject—which are a common symptom of mental health disorders (Kirstein and Smith 1980). Lastly, the lily questions its own nature. In modern psychology and studies of personality, such a phenomenon is known as self-doubt, which affects our well-being by challenging the foundations of our self-conception (Braslow et al. 2012; Zhao and Gong 2019; Zhao et al. 2019).

These negative mental and physical symptoms are caused by the lily's acting against its nature. This Aristotelian–Thomistic *ad naturam* argument is the key to Kierkegaard's position on the harmful effect of social comparison on our well-being. In his *Nicomachean Ethics*, Aristotle argues that the good of a given thing is connected to its proper function and the excellent performance of that function. He states in *Nicomachean Ethics*: "just as the good, i.e., [doing] well, for a flautist, a sculptor, and every craftsman, and in general, for whatever has a function and [characteristic] action, seems to depend on its function, the same seems to be true for a human being, if a human being has some function" (1097b25-30). For Aristotle, the appropriate function of a human being is rational activity in accord with virtue, and the excellent performance of that proper function is using reason in various situations. The problem that one identifies in the lily's behavior is that it did not act according to its appropriate function, but acted as if it were a different species. The essential "whatness" of the lily is being compromised by its preoccupation, ambition, emotive states, and behavior.

Kierkegaard projects the generally conceived harmful dimension of social comparison onto the moral-religious sphere of human life. The argument concerning the frustration of human nature is married with Kierkegaard's perfectionist view of the human self. The disposition to compare with others is a vice of character for three reasons. First, in social comparison, we become addicted to pleasure that is either parasitical or pathological. The former we derive from downward comparison; the latter we derive from upward comparison. Second, the life of social comparison is a form of harmful habituation that affects

our moral capacity. More specifically, the more habituated we are in social comparison, the less freedom we possess to exercise our moral sensitivity, moral judgment, and moral motivation. Kierkegaard points out our comparative habits in our disjointed identifying of the tax collector as our model while habitually adhering to the principle of social comparison. Third, social comparison is also horizontal (Locke 2005). It is a regulative form of comparison that allows us to stay within boundaries of social rules espoused by our peers and enhances group harmony (Baldwin and Mussweiler 2018). This type of social comparison has, for Kierkegaard, the most deindividualizing power over an individual. It relates to our inclination to seek pleasure, habituate, and gather in groups in the process of socialization.

Alas, as one grows older, one gradually becomes habituated to a great deal in life. Among other things, one also becomes accustomed to saying a great deal that one does not really mean; among contemporaries one becomes accustomed to talking within so many presuppositions that the simple and the lofty things almost sink into oblivion. . .. Alas, one so easily becomes habituated in life, in habit's dull round of association with others, to the point of almost abandoning oneself while one plays with platitudes (UDVS: 57–58).

Elsewhere, Kierkegaard implies: "To bring about similarity among people in the world, to apportion to people, if possible equally, the conditions of temporality, is indeed something that preoccupies worldliness to a high degree" (WL: 71). Speaking of the deindividualizing phenomenon of leveling in *Two Ages*, Kierkegaard states: "A demon that no individual can control is conjured up, and although the individual selfishly enjoys the abstraction during the brief moment of pleasure in the leveling, he is also underwriting his own downfall" (TA: 86).

While all forms of social comparison are harmful to the ideal of a single individual existence for Kierkegaard, horizonal social comparison lowers the bar for what is morally and religiously required from every human being. In *The Sickness unto Death*, "the single individual" represents a normative requirement for religious existence as a human being becomes a singular sinner as the single individual. Without the category of the single individual (a philosophical/moral self), there is no category of the ideal Christian (a Christian self) (SUD: 119; JP 2: 1781). This equation between the moral and religious makes social comparison into a moral-religious vice for Kierkegaard.

## 5. Conclusions

Kierkegaard's formulation and analysis of the "worldly life of comparisons" offers an insightful perspective on the phenomenon of social comparison, nowadays widely analyzed in the sciences, *avant la lettre.* We learn from his writings that social comparison, while a biological and environmental endowment, is harmful to us as individual human beings that function in modern society. Social comparison is a vice, and how Kierkegaard presents it sheds light on the intertwined nature of the non-moral and non-religious and the moral-religious. The harm caused by social comparison pertains to our overall mental and physical well-being. While part of our nature, unchecked by rational reflection and will, social comparison gets us to act against our nature, for Kierkegaard. It is a vice that affects us below our cognitive register. Social comparison is habituated. To combat social comparison, we need to understand its mechanics, meaning its moral psychology, and think less about ourselves and more about others (Lippitt 2019). Social comparison is a deindividualizing vice that has a significant bearing on our moral and religious life. It is so inasmuch as, for Kierkegaard, to be a Christian, one must first and foremost be a single individual. To fight social comparison, we must cultivate the virtue of humility and, as Kierkegaard adds, the virtue of reliance on God.

**Funding:** This research was funded by the National Science Centre (Poland) and the European Union's Horizon 2020: Marie Skłodowska-Curie, grant agreement No. 945339, grant number 2022/47/P/HS1/01942.

**Institutional Review Board Statement:** Not applicable.

**Informed Consent Statement:** Not applicable.

**Conflicts of Interest:** The authors declare no conflict of interest.

## Notes

[1]    All references to Kierkegaard's texts (Kierkegaard 1967–1978, 1978, 1980, 1993, 1995, 1997, 1998) will be made parenthetically as above to the appropriate English translation, using a widely used sigla, as noted in the References section.

[2]    This intuition about the negative influence of both types of comparison has been confirmed in many studies on social comparison: See: (Buunk et al. 1990).

[3]    Interestingly, at times, Kierkegaard makes claims that pertain to his perceived greater suffering against those of other people ("No, even though compared with men generally I can be said to have suffered unusually"; JP2:1812; Cf. JP 6: 6906).

[4]    A life of comparison is, to use Heidegger's conceptual apparatus, a way of being in the world. A life of comparison is, to use Iris Murdoch's vocabulary, a way of seeing oneself and others.

[5]    https://www.nhs.uk/mental-health/conditions/depression-in-adults/symptoms/, Access date: 22 September 2023.

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
