# Peer review of "The Vice of Social Comparison in Kierkegaard: Nature, Religious Moral Psychology, and Normativity"

_religions, doi:10.3390/rel14111394_

Round 1

Reviewer 1 Report

Comments and Suggestions for Authors

     In general, the essay as considerable merit. It draws attention to the importance of the problematic nature of social comparison in Kierkegaard's literature. It exhibits a careful reading of Kierkegaard's texts and a thorough familiarity with the secondary literature, building particularly on the work of John Davenport and John Lippitt (but not uncritically). The most significant portions of the essay are its analyses of the dangers of social comparison according to Kierkegaard and its exacerbation by the social dynamics of modernity.   

    I do have three questions about the essay. First, it implies that Kierkegaard does admit that some social comparison is necessary for any cohesive social life. I wish that the author could provide more textual evidence from Kierkegaard's authorship that he did have a qualified appreciation for the need for benign social comparison.

      Some of the argument that Kierkegaard has a naturalistic, non-moral, and non-religious dimension to view of human teleology relies on the notion that unrestricted social comparison is detrimental to our flourishing. But Kierkegaard seems to identify the telos of life with virtues that include contruing all life, and one's position in it, as a gift to be received with humility and gratitude. That sounds like a religious dimension. Perhaps the distinction between "eudaimonism" and the telos of "eternal happiness" does not hold for Kierkegaard, which would undermine the claim about "non-religious naturalism. The author could say more about this.

     Finally, sometimes the author seems to elide "social comparison" with any kind of comparison, such as the comparison of worldly happiness with eternal happiness. 

     In spite of these issues, the essay is provocative, significant, and well-worth publishing.

"

Author Response

Dear Reviewer 1,

Thank you kindly for this thorough review of my paper. Please see below my responses to your remarks:

Reviewer 1 -Response

  1. Good remark: I have added a few clarificatory sentences with references that draw a distinction between comparisons that are social/moral/religious (+other values) and yet somehow admissible and social comparison proper that Kierkegaard vehemently criticizes.

Some forms of comparison that Kierkegaard admits (and at times recommends) have a social and moral-religious dimension. Throughout his Journals Kierkegaard states that Christ (“the God-man as the prototype”) is “the criterion for being human,” and we should make conscious comparisons with Him to understand how far we are from the ideal of human being represented by Christ (JP 2: 1802; JP 1: 334; JP 2: 1895). We are also invited to compare between people to have a good grasp of important qualities that are both religious and non-religious, such as dishonesty, mediocrity, but also genius (Cf. JP 3: 3088; JP6: 6553; JP6: 6560). However, these comparisons are not forms of social comparison in the sense that Kierkegaard criticizes in his writings. Rather than comparing oneself with particular individuals or groups we should compare and conform our lives to the standard of existence that is “all men” (JP3: 2966).

Added note: Interestingly, at times Kierkegaard makes claims that pertain to his perceived greater suffering against those of other people (“No, even though compared with men generally I can be said to have suffered unusually”; JP2:1812; Cf. JP 6: 6906).

  1. I struggle to understand this second remark. It is brief and complex, at the same time. The reviewer kindly suggests: “Some of the argument that Kierkegaard has a naturalistic, non-moral, and non-religious dimension to view of human teleology relies on the notion that unrestricted social comparison is detrimental to our flourishing.” Sincerely, I am not sure exactly which of the “some of the argument” is the culprit here. I fear that I may have not communicated my position clearly enough to perhaps confuse the reviewer. I have tried to show that my reading of the natural argument for human flourishing in Kierkegaard is important, yet it does not take precedence over his religious teleology. I am in fact complementing the religious perspective with the non-religious, with a view that the latter (while valid on its own terms), serves the former. I think functionalism is clearly evident in SK’s writings; but functionalism cannot, so to speak, save us. I think the distinction between eudaimonism and authentic existence is too sharp, yet the distinction between eudaimonism and “eternal happiness” stands! This is so as SK does not clearly develop eudaimonism to the degree he does so with “eternal happiness.” I will tweak some wording to this effect.
  2. This manuscript indeed requires a stronger systematization of when comparison is a mental function, perhaps, non-moral, and when social comparison is at stake. I have amended several fragments of my manuscript to that effect.

Reviewer 2 Report

Comments and Suggestions for Authors

Great manuscript! I'm grateful for the chance to review it. I find the argument compelling and thorough with respect to Kierkegaard and the virtues/vices of comparison and imitation.

I have only two main suggestions.

1. The first is merely formatting. After block quotations, there is an indent in the body every time. Are all of them genuinely a new paragraph? Or are some of the statements following block quotations elaborations on the block quotation that technically belong to the same paragraph?

2. The second issue I have is some concern about subtle anti-Judaism in the discussion of the Pharisee and tax-collector parable (lines 335-367). I think it would be best to consult Amy-Jill Levine's "Short Stories by Jesus" on that parable in order to make sure to avoid any hint of anti-Judaism. For example, at lines 355-356 where it says it takes minimal effort to identify the Pharisee as the "bad character." Is that how the original Palestinian Jewish audience would have taken it? Pharisees were awesome (like Hillel!), so it would have created a lot of cognitive dissonance with the original audience--given (as this manuscript itself says) that the tax-collector is a Roman collaborator. It would be quite difficult to identify with the tax-collector, and to see any flaws with the Pharisee. Even if the anti-Jewish or anti-Pharisee reading itself is from Kierkegaard and not the current author, it would be best to note this at least briefly.

Author Response

Thank you, Reviewer 2, for the kind words and insightful remarks. 

  1. I have revisited the indentations. They should look better now. The final pair of eyes, the typesetter, will make sure all is good in that respect. 
  2. This is a helpful remark. I have added a note to signal that the perspective in the tax-collector/Pharisee parable is clearly Christian and readily available to Kierkegaard's contemporaries. It is rather uncontroversial that Pharisees were figures that were admired for their high moral standards, piety, and exemplary "academic/scientific" work. I hope that these edits will assure that Kierkegaard's perspective on Jesus's parable is not a historically accurate rendering of the society in Judea in Jesus' times. 

Edited sentences include: 

    1. "It is not the Pharisee, a widely respected figure in Jesus’s times for their high moral standards, piety, and scholarly work. This is so because, as the story tells, the Pharisee justified himself before God."
    2. "Realizing that the Pharisee is the bad character in the told parable requires minimal effort for Kierkegaard’s contemporaries, but also for contemporary readers."